# Formal Meta-Analysis of Hypoxic Gene Expression Profiles Reveals a Universal Gene Signature

**DOI:** 10.3390/biomedicines10092229

**Published:** 2022-09-08

**Authors:** Laura Puente-Santamaria, Lucia Sanchez-Gonzalez, Nuria Pescador, Oscar Martinez-Costa, Ricardo Ramos-Ruiz, Luis del Peso

**Affiliations:** 1Departamento de Bioquímica, Instituto de Investigaciones Biomédicas “Alberto Sols” (CSIC-UAM), Universidad Autónoma de Madrid (UAM), 28029 Madrid, Spain; 2Genomics Unit Cantoblanco, Fundación Parque Científico de Madrid, C/Faraday 7, 28049 Madrid, Spain; 3Centro de Investigación Biomédica en Red de Diabetes y Enfermedades Metabólicas Asociadas (CIBERDEM), Instituto de Salud Carlos III, 28029 Madrid, Spain; 4IdiPaz, Instituto de Investigación, Sanitaria del Hospital Universitario La Paz, 28029 Madrid, Spain; 5Centro de Investigación Biomédica en Red de Enfermedades Respiratorias (CIBERES), Instituto de Salud Carlos III, 28029 Madrid, Spain; 6Unidad Asociada de Biomedicina CSIC-UCLM, 02006 Albacete, Spain

**Keywords:** transcription, hypoxia, RNAseq, meta-analysis

## Abstract

Integrating transcriptional profiles results in identifying gene expression signatures that are more robust than those obtained for individual datasets. However, a direct comparison of datasets derived from heterogeneous experimental conditions is problematic, hence their integration requires applying of specific meta-analysis techniques. The transcriptional response to hypoxia has been the focus of intense research due to its central role in tissue homeostasis and prevalent diseases. Accordingly, many studies have determined the gene expression profile of hypoxic cells. Yet, despite this wealth of information, little effort has been made to integrate these datasets to produce a robust hypoxic signature. We applied a formal meta-analysis procedure to datasets comprising 430 RNA-seq samples from 43 individual studies including 34 different cell types, to derive a pooled estimate of the effect of hypoxia on gene expression in human cell lines grown ingin vitro. This approach revealed that a large proportion of the transcriptome is significantly regulated by hypoxia (8556 out of 20,888 genes identified across studies). However, only a small fraction of the differentially expressed genes (1265 genes, 15%) show an effect size that, according to comparisons to gene pathways known to be regulated by hypoxia, is likely to be biologically relevant. By focusing on genes ubiquitously expressed, we identified a signature of 291 genes robustly and consistently regulated by hypoxia. Overall, we have developed a robust gene signature that characterizes the transcriptomic response of human cell lines exposed to hypoxia in vitro by applying a formal meta-analysis to gene expression profiles.

## 1. Introduction

Oxygen homeostasis is essential to sustain cellular metabolism in eukaryotes. Hypoxia triggers multiple adaptive mechanisms, from metabolism reprogramming to tissue restructuring, aimed to re-balancing oxygen supply and demand [1]. In multicellular organisms this response can be very diverse, depending on cell type, extension and degree of the oxygen deprivation, or pathological state.

Most of these responses are orchestrated at the transcriptional level, with the Hypoxia Inducible Factors (HIFs) being the main drivers of the hypoxic gene expression pattern [2]. The heterodimeric HIF transcription factor consists on a β subunit (ARNT), constitutively expressed, and an α subunit (HIF1A, EPAS1, HIF3A) which, in normoxic conditions, is marked for degradation by the concerted action of a family of oxygen-dependent enzymes (EGLN family) and the von Hippel-Lindau (VHL) ubiquitylation complex [3,4,5]. When oxygen concentration decreases, the α subunits escape degradation due to the reduced activity of the EGLNs, translocate to the nucleus and bind to Hypoxia Response Elements along the β subunit. Transcriptional activity of HIFs depends also on interaction with co-activators such as CREB-binding protein or p300, whose binding is also regulated in an oxygen-dependent manner [6,7].

Given the importance of the transcriptional response for tissue oxygen homeostasis and its alteration in disease, a large number of works have attempted to identify the full set of genes regulated by hypoxia through gene profiling experiments. Since these studies were performed in a wide variety of experimental conditions (cell types, oxygen tension, exposure time) integrating their results could lead to identify a set of genes ubiquitously regulated by hypoxia, as well as genes whose alteration is restricted to specific situations in addition to hypoxia. However, little effort has been done in this regard and, to the best of our knowledge, only two attempts to integrate all the hypoxic gene profiling experiments have been done [8,9]. The first analysis of this type, based on the analysis of gene profiles generated by means of DNA microarrays, produced the first list of genes universally induced by hypoxia and revealed that the set of genes induced by hypoxia were more conserved than those repressed [8]. A second, more recent study, exploited the information derived from RNA-seq experiments producing a more comprehensive list of hypoxia-regulated genes and characterized HIF-isoform common and specific targets [9]. In spite of their merit, none of these works employed formal meta-analysis approach for their analysis which, given the heterogeneous nature of the data, is critical to draw statistically sound conclusions [10].

Among the various meta-analysis methods applicable to transcriptomic data [11], we employed a model that combines the effect sizes (log fold change of the ratio of the gene expression under hypoxia and normoxia) on gene expression across the individual studies. This allowed us, not only to identify the set of genes differentially regulated in response to hypoxia, but also to estimate the magnitude of change for each gene. Instead of assuming a fixed effect of hypoxia on any given gene across the different studies, we used a random effects model that considers that the true effect could vary from study to study to reflect, for example, the different response in distinct cell types. In this study we aim to define core components of the transcriptional response to hypoxia taking advantage of the wider public availability of next generation sequencing data, RNA-seq in particular. Applying a random effects model to the expression data gathered we were able to define a molecular signature representing the early (≤48 h) transcriptional response to hypoxia, independently of cell type.

## 2. Materials and Methods

### 2.1. RNA-seq Data Download and Processing

Raw reads of the RNA-seq experiments were downloaded from Sequence Read Archive [12]. Pseudocounts for each gene were obtained with salmon [13] using RefSeq [14] mRNA sequences for human genome assembly GRCh38/hg38 as reference.

Differential expression in individual subsets was calculated with the R package DESeq2 [15] using local dispersion fit and apeglm [16] method for effect size shrinkage.

### 2.2. Meta-Analysis

The meta-analysis intended to identify the effect of sustained hypoxia on early gene expression in human cells compared to normoxic controls. To identify studies to be included in the meta-analysis Gene Expression Ommibus (GEO) repository was searched with the terms ‘hypoxia[Description] AND “expression profiling by high throughput sequencing” [DataSet Type]’ on 11 February 2021. The search resulted in a total of 394 studies. We only kept studies performed in human cells that determined steady-state RNA levels in total (poly-A) RNA samples and excluded analysis that did not include replicates, employed treatments other than reduced oxygen tension (e.g., chemical inhibitors or other hypoxia mimetics) or those where gene expression was analyzed after 48 h. We also excluded studies that used cycling/intermittent hypoxia or that were performed in non-human cell lines. A total of 46 studies (independent GSE entries) remained after application of the inclusion/exclusion criteria and were used for the meta-analyses (Appendix A). A pooled estimate of the size effect of hypoxia on expression was determined for each gene using the R packages metafor [17] and meta [18] using as input the log2-Fold change value and its associated standard error computed for each individual RNA-seq experiment using the R package DESeq2 [15]. Given that the individual estimates derive from an heterogeneous group of experiments, including different cell types and experimental conditions, we assumed that these individual estimates derive from a distribution of true effect sizes rather than a single one and thus applied a random-effects model for the meta-analysis. Since some of the selected studies included several cell types and/or experimental conditions (see results for details), we fitted a 3-level model [19] that, in addition to sampling error and between-study heterogeneity, takes into account possible dependencies between data subsets derived from a single study.

The meta-analysis intended to identify the effect of sustained hypoxia on early gene expression in human cells compared to normoxic controls. To identify studies to be included in the meta-analysis Gene Expression Ommibus (GEO) repository was searched with the terms ‘hypoxia[Description] AND “expression profiling by high throughput sequencing” [DataSet Type]’ on 11 February 2021. The search resulted in a total of 394 studies. We only kept studies performed in human cells that determined steady-state RNA levels in total (poly-A) RNA samples and excluded analysis that did not include replicates, employed treatments other than reduced oxygen tension (e.g., chemical inhibitors or other hypoxia mimetics) or those where gene expression was analyzed after 48 h. We also excluded studies that used cycling/intermittent hypoxia or that were performed in non-human cell lines. A total of 46 studies (independent GSE entries) remained after application of the inclusion/exclusion criteria and were used for the meta-analyses (Appendix A). A pooled estimate of the size effect of hypoxia on expression was determined for each gene using the R packages metafor [17] and meta [18] using as input the log2-Fold change value and its associated standard error computed for each individual RNA-seq experiment using the R package DESeq2 [15]. Given that the individual estimates derive from an heterogeneous group of experiments, including different cell types and experimental conditions, we assumed that these individual estimates derive from a distribution of true effect sizes rather than a single one and thus applied a random-effects model for the meta-analysis. Since some of the selected studies included several cell types and/or experimental conditions (see results for details), we fitted a 3-level model [19] that, in addition to sampling error and between-study heterogeneity, takes into account possible dependencies between data subsets derived from a single study.

### 2.3. Functional Enrichment Analysis

Enrichment of Gene Ontology terms was performed with the Bioconductor’s clusterProfiler package [20] using a q cut-off value of 0.05. The list of background genes included those expressed in at least 90% of the datasets and as foreground list the subset of genes significantly regulated by hypoxia (*FDR* < 0.01) with $abs(Log_{2}FC)\geq 0.7$. As background genes. To reduce the redundancy, highly similar GO terms were removed keeping a single representative by using the “simplify” function using a cut-off value of 0.6 (up-regulated genes) or 0.7 (down-regulated genes). The much larger number of enriched terms found for up-regulated genes justified the use of a slightly more lenient cutoff value for the simplify function. Gene Set Enrichment Analysis [21] was performed using the preranked tool of the Broad Institute’s application for Linux (version v4.2.3). The Canonical pathways subset was used as gene set database and gene list was ranked according to the pooled LFC estimate derived from the meta-analysis. Genes expressed in less than 5% of the studies’ subsets were removed from the list. Pathways with an FDR<0.01 were considered significantly enriched.

## 3. Results

### 3.1. Hypoxia-Induced Transcriptional Profiles Show Limited Overlap

In order to identify genes consistently regulated by hypoxia across a wide range of cell types and experimental conditions, we compared the results from 46 studies analyzing the transcriptional response to hypoxia by means of RNA-seq (Appendix A). Since some studies included several cell types, oxygen tensions or times of exposure to hypoxia, we took subsets of the study’s data so that each one included a single cell line and set of experimental conditions (Figure 1). Thus, our initial data set included a total of 81 subsets of normoxia-hypoxia paired samples, each one comprising a single cell line, exposure time and oxygen tension (Table 1).

For each of these 81 subsets, we identified the genes significantly regulated (FDR<0.01) in response to hypoxia and recorded the number of times each gene was found to be down- or up-regulated across the 81 subsets (Figure 1A). This analysis revealed that the majority of genes showed significant changes in a small number of datasets (Figure 1B). Thus, of a total of 15,362 genes found significantly (FDR<0.01) repressed by hypoxia across all the 81 analyzed datasets, over 50% of them were found in a maximum of five datasets (Figure 1B, first red bar). Similarly, a total of 16,872 genes were significantly (FDR<0.01) induced in at least one dataset, but 60% of them were shared in a maximum of five datasets (Figure 1B, first blue bar). Conversely, not a single gene was found consistently down- or up-regulated across all the datasets, but induced genes tend to be more consistently regulated (Figure 1B). The most frequently repressed genes were present in at most 50–55 datasets while several genes were found significantly up-regulated in 65–70 of them (Figure 1B and Appendix A). These results suggested a reduced overlap of the analyzed transcriptional profiles. Next, we performed all possible pair-wise comparisons between the 81 lists of DEGs (Figure 1A, bottom right panel). As shown in Figure 1C, the overlap between lists of DEGs was below 10% in the vast majority of pair-wise comparisons with a median value of 3.6% of shared genes between lists of repressed genes and a median value of 6.8% in the case of the induced genes. Altogether these results indicate a considerable heterogeneity in the transcriptional response to hypoxia, which is more pronounced in the case of repressed genes.

### 3.2. Identification of Robust Transcriptional Responses to Hypoxia

Given the heterogeneity in the transcriptional response to hypoxia, we decided to apply a formal meta-analysis to identify genes significantly regulated by hypoxia across all the datasets and estimate the magnitude of the change in their expression. To this end, for each of these 81 subsets we estimated the difference in expression levels between normoxic and hypoxic conditions for all genes (“LFC”, Figure 2). From these analyses we extracted the statistics (Effect size, “LFC”, and standard error associated to this estimate, “SE”) for individual genes across the different studies and performed a meta-analysis on each one of these gene-specific datasets to estimate the pooled effect of hypoxia (Figure 2 Meta-analysis). Thus, an independent meta-analysis was performed for each individual gene by integrating the effects of hypoxia on that particular gene across the different studies and conditions. The results provide a pooled estimate of the effect of hypoxia on the expression of the gene under analysis and its statistical significance. As an example, the results of the meta-analysis for the EGLN3 gene, encoding a cellular oxygen sensor known to be directly regulated by HIF in response to hypoxia [22], are shown in Appendix A. Finally, we compiled the pooled estimates for all genes detected in more than one subset, together with the statistical significance value, to produce a table representing the overall effect of hypoxia on gene expression (Figure 2 “Compiled MA results”).

The quality of the meta-analyses’ results is critically dependent on the original data fed to the model. In this regard, correlation analyses revealed a few incoherent datasets (Appendix A). In those cases where the lack of positive correlation was clearly due to a mistake in the labeling of samples in public databases, as indicated by a large negative correlation coefficients (Appendix A, subsets “S42” and “S58”), the treatment labels were correctly set and the study was kept. The remaining incoherent studies, having a correlation coefficient not significantly different to zero (p>0.01), were discarded. After these data sanity check procedures, the whole analysis strategy (Figure 2) was repeated on this corrected and filtered dataset. Table 1 shows the statistics of the data set after filtering and Appendix A, the full description of each of the samples included in the final analysis.

### 3.3. Identification of a Universal Core of Hypoxia-Inducible Genes

The results of the meta-analysis on the clean dataset, after filtering out the outlier subsets and removing genes detected in less than 5% of the subsets, revealed 6242 genes (out of a total of 20,918) whose expression was significantly (FDR<0.01) altered in response to hypoxia (Figure 3A and Appendix A), with similar number of genes being induced (3043) and repressed (3199). These numbers are larger than the typical values obtained in individual experiments, with median values of 1294 and 1442 genes significantly down- and up-regulated respectively (Figure 3B). The large number of DEGs identified by the meta-analyses is probably a consequence of the increased power to detect small effect sizes due to the integration of a large number of samples. In agreement, the median effect size (LFC) observed for the genes differentially expressed (DE) according to the meta-analyses are −0.31 and 0.42 for down- and up-regulated genes respectively, contrasting with the median effect size observed in individual studies of −0.76 and 0.86 for down- and up-regulated genes respectively (Figure 3C). Accordingly, the identification of DEGs based only on statistical significance yields a large number of genes barely changing in response to hypoxia (Figure 3A, genes labelled “*FDR*” in blue colour). As an example, the smallest effect size found among significantly up-regulated genes is 0.11 corresponding to fold induction over normoxia of about 1.1 times.

In view of these results we tried to identify a minimum effect size likely to represent biologically relevant changes in gene expression. To this end we explored the relationship between effect size and belonging to biological processes known to be regulated by hypoxia, testing whether increasing the Log_2_FC cut-off would also increase the proportion of genes with hypoxia related annotations. As shown in Appendix A, the *p*-value for the association of biological function and regulation by hypoxia reached a minimum at effect size (Log_2_FC) values between 0.3 and 1.7. Since the choice of effect size values only affects the distribution of genes into categories (i.e., differentially expressed versus not altered) but does not change their total number, the minimum *p*-value corresponds to the least likely distribution expected by chance. Thus, we decided to take these values as the lower boundary required to produce a biological response to hypoxia. The median value of the effect sizes is 0.7, corresponding to an induction of 1.6 times over basal levels (0.6 times the normoxic level for repressed genes).

Thus, in response to hypoxia a total of 926 genes, 167 repressed and 759 induced, show a statistically significant change in expression (FDR<0.01) of a magnitude likely to be biologically meaningful ($|Log_{2}FC|>0.7$) (Figure 3A labeled in green and red colours).

The difference in the number of repressed and induced genes is a consequence of the distribution of effect size values having a longer tail in the latter case (rug plot of the x-axis in Figure 3A,D left panel). The different shapes of the distribution of effect size values also suggest that hypoxia has a relatively weak effect on gene repression. In fact, while the number of significantly induced genes is about four times higher that of repressed genes (759 vs. 167) for an effect size higher than 0.7, the ratio increases to seventeen times more up-regulated than down-regulated genes (424 vs. 25) for effect sizes larger than 1. Since the meta-analyses included experiments done at relatively short exposure times (27% of the subsets correspond to exposure times ranging from 1–12 h), it could be argued that the smaller effect size observed for repressed genes is a consequence of short-time experiments failing to detect the effect on mRNA levels due to the relatively long half-life of mRNAs. To test this hypothesis, we repeated the meta-analyses selecting only subsets corresponding to treatments of 24–48 h, significantly longer than the median half-life of 5.7 h observed for human mRNAs under hypoxia [23]. As shown in Figure 3D left panel, both distributions show a small shift toward higher absolute effect size values, but the difference between them remains unaltered. Thus, the relatively smaller effect of hypoxia on gene repression does not appear to be due to the persistence of mRNA molecules present prior hypoxia exposure.

Finally, in order to get a list of core hypoxia-responsive genes we identified those that were ubiquitously expressed. To that end, we selected those genes whose expression, averaged across conditions, was detectable in at least 90% of the analyzed subsets (Figure 3A labeled in red color). The resulting list included a total of 295 genes (114 down- and and 181 up-regulated) consistently altered by hypoxia across conditions. These genes correspond to those most frequently found significantly regulated across individual datasets (Appendix A). The top 5 most frequently down- and up-regulated genes are labelled in Figure 3A.

Functional enrichment of Gene Ontology terms, indicated that core hypoxia-induced genes are mainly involved metabolic reprogramming but also in differentiation and morphogenesis, being the development of the circulatory system particularly prominent (Figure 4A). On the other hand, the genes consistently repressed by hypoxia across conditions, are involved in cell cycle progression, DNA replication and repair, ribosome/rRNA biogenesis and metabolism of amino acids (Figure 4B). Similar results were obtained by Gene Set Enrichment Analysis (GSEA) using the meta-analysis derived LFC pooled estimates as ranking factor and pathway databases (Biocarta, KEGG, PID, Reactome and Wikipathways) as source of gene sets [24]. GSEA results showed that cell cycle, DNA replication and DNA repair pathways were repressed by hypoxia (Appendix A). In addition, mitochondrial respiratory electron transport and complex I biogenesis were also found repressed (Appendix A). On the other hand, HIF-related pathways and glucose metabolism were upregulated by hypoxia (Appendix A).

In summary, the application of a formal meta-analysis to hypoxia gene expression profiles using a random effects model lead to the identification of a core set of 295 ubiquitously expressed genes whose expression is significantly altered by hypoxia by a factor of at least 0.7 log_2_-units. The identity of these genes and along their response to hypoxia across individual subsets can be found in Appendix A.

### 3.4. Consistency of Meta-Analysis Results

To test the consistency of the pooled estimates described above, we applied a leave-one-out cross-validation, a common method to estimate how accurately a predictive model will perform on new data. To this end, we performed a set of meta-analyses using as input all data subsets except for one and then compared the estimated effect sizes with the actual LFC observed in the subset that was left out (Figure 5A). The process was repeated until all possibilities were exhausted. This approach yielded a list of 70 correlation coefficients corresponding to each iteration. As shown in Figure 5B, in almost all cases there was a strong correlation between the pooled estimates and the actual effect sizes observed in the individual subset that was left out of the meta-analyses, with 50% of the instances showing a Pearson’s correlation coefficient over 0.81 and 75% of the cases above 0.72. We also analyzed the overlap between the DEG derived from each meta-analyses and those from the individual experiment that was left out from it and found a median value of 19% percent of shared genes between lists of repressed genes and a median value of 18% percent in the case of the induced genes (Figure 5B). These values contrast with the low overlap found in pairwise comparisons between individual experiments (Figure 1C), in particular in the case of down-regulated genes. Finally, we analyzed the percentage of core genes genes (FDR<0.01, $|Log_{2}|>0.7$ and present in at least 90% of the subsets included in the meta-analysis) that were present in the DEG (FDR<0.01) from the subset not included in the meta-analysis. This analysis showed than core genes are consistently found among the DEG identified individual studies, with median values of 55% of the core repressed genes and 65% of the core induced genes (Figure 1D). Altogether these results indicate that the ensemble of pooled estimates predict with high accuracy the response to hypoxia and the identity of DEGs in new experiments not included in the meta-analysis.

### 3.5. Comparison of Meta-Analyses Results with a Reference Hypoxia Signature

The core set of genes identified in the analyses described before can be considered a signature of the transcriptional response to hypoxia. Thus, we next compared the core of hypoxia-inducible genes derived from the meta-analysis with the MSigDB’s Hallmark hypoxia geneset [24], a widely used gene signature composed of 200 genes up-regulated in response to low oxygen levels. As shown in Figure 6A, the overlap between both gene sets was relatively small, with less than one third (64 out of 200) of the genes in the Hallmark hypoxia signature being present in the meta-analyses derived geneset, in spite of both genesets being similar in size, median Log_2_FC and nearly universal expression (Appendix A). Moreover, the overlap was only moderately increased when the Hallmark hypoxia signature was compared to the geneset derived from the meta-analysis without restricting to ubiquitously expressed genes (Figure 6B). In order to understand the cause for the reduced overlap, we analyzed the effect of hypoxia on the expression of the 109 genes present in the Hallmark hypoxia signature only (Appendix A). Five of the genes in this group (*CCN5*, *CCN1*, *BRS3*, *CCN2* and *LALBA*), were not among the 22,182 genes considered in the meta-analyses, probably due to the lack of detectable expression in the RNA-seq datasets. The effect of hypoxia on the remaining 104 genes is shown in Figure 6C. This analysis revealed that 40% percent of these genes (43 out of 104) were not present in the meta-analyses-derived signature because the pooled estimate of their induction by hypoxia was below the threshold value of 0.7, in spite of being statistically significant (labeled as “DEG_UP” and shown in green color). However, the remaining 60% of the genes (62 out of 104) did not show a statistically significant induction by hypoxia (57 out of 104; Figure 6C labeled as “Non_DEG” and shown in red color) or were repressed (5 out of 104 genes; Figure 6C labeled as “DEG_DN” and shown in blue color). A forest plot representing the pooled estimate of the LFC for the genes labeled as “Non_DEG” or “DEG_DN” shows that they cluster around the value of zero and that, in many cases, confidence interval for the point estimate is wide (Figure 6D) and overlaps the value of zero (no regulation). These results indicate that these 62 genes are either not consistently induced by hypoxia or show a cell-type/condition specific induction. An example of the latter is the HMOX1 whose expression is induced, repressed or left unaltered depending on the cell-type and/or experimental conditions (Appendix A). Interestingly, among the genes whose regulation by hypoxia depends on the specific experimental conditions are *ALDOB*, and *LDHC*, which are tissue-specific paralogs of genes strongly and consistently induced by hypoxia, *ALDOA* and *LDHA* respectively (Figure 6D, labelled in red).

Altogether, these results suggest that the meta-analyses derived gene signature improves the gene sets derived from individual studies by excluding genes whose regulation is cell type or condition specific and those with effect sizes of small magnitude.

## 4. Discussion

The integration of multiple datasets representing the transcriptional response to a given stimulus, allows for the identification of consistent changes in gene expression. However, transcriptional profiles are noisy, and the correlation between them is poor [25,26]. Thus, the number of common DEGs decreases rapidly with the number of studies taken into consideration (Figure 1). To identify genes commonly regulated by hypoxia one can set a minimum number of studies where the gene needs to be found as a DEG [9]. Then again, there is no objective criteria to select minimal thresholds and this approach results in a list of commonly regulated genes which does not provide information regarding the magnitude of their regulation. Fortunately, applying meta-analysis methods appears to be a good and practical solution to reduce noise and increase signal across different studies [10].

Herein we describe the application of a formal meta-analysis procedure to identify genes whose expression is significantly modulated across a number of different gene profiling studies. This approach not only provides the identity of the genes but also a pooled estimate of the effect of the condition on the expression. Moreover, by applying a random effects model, this strategy takes into account the wide variability in gene expression expected from the integration of transcriptomes derived from different experimental conditions. The application of this approach to 70 paired normoxic/hypoxic transcriptomes representing a total of 430 samples resulted in the identification of 6242 genes, roughly 30% of the detectable genes, as significantly (FDR<0.01) regulated in response to changes in oxygen tension. These results beg answering the question of the biological relevance of statistically significant but small changes in gene expression. For example, the median effect size for significantly up-regulated genes was 0.42, corresponding to a fold induction of 1.34 times over normoxia. Thus, for half of the significantly-induced genes, the level of mRNA in hypoxia is at most 1.34 times higher than control levels.

For some genes this small increase in expression could have important consequences, but statistical significance by itself does not warrant biological relevance. For this reason we sought to identify an effect size that it is likely to have an impact of cellular biochemistry. To this end we recorded the changes in expression of genes known to have an impact on different biological processes upon exposure to hypoxia and took the median effect size value, 0.7 log_2_ units, as threshold. As only a fraction of the genes in a category are induced to initiate the biological response, this value is likely to be an underestimation and thus could be considered a lower boundary to identify biologically relevant changes. The importance of considering the effect size (Log_2_FC), and not only the statistical significance to identify DEGs, is a strong argument in favor of formal meta-analysis instead of other integrative methods that yield a consensus list of genes without an associated estimate of the effect of hypoxia. The effect of hypoxia on gene repression is apparently weaker than on gene induction, as indicated by relative smaller number of repressed genes above the effect size threshold. This could be a consequence of repression being indirectly regulated by HIFs [8,27,28] and would explain the higher heterogeneity of the response observed for repressed genes (Figure 1). The indirect regulation could occur via transcriptional regulators acting downstream of HIF [29] or be a consequence of the cellular adaptations to hypoxia. For example, alteration of energy availability during hypoxia could prevent cell division with the concomitant down-regulation of genes involved in DNA synthesis and cell cycle progression. Thus, further work is required fully understand the mechanisms responsible for gene repression during hypoxia.

As a further advantage, the application of a formal meta-analysis approach allows for the application of all associated statistical tests, including moderator analysis, to study the effect of different factors on the regulation of gene expression. Through the application of this analysis we found that endothelial cells are deficient in the induction of a relatively large set of genes in response to hypoxia. Among those genes there are many enzymes involved in glucose metabolism, particularly in glycolysis and synthesis of glycogen (data not shown). It is known that HIF1A, but not EPAS1, is responsible for the hypoxic induction of glycolytic genes [30], it is tempting to speculate that the specific pattern of expression observed in endothelial cells could be a consequence of the relative importance of EPAS1 over HIF1A isoform in this cell type. However, since most endothelial datasets consist of experiments performed on Human Umbilical Vein Endothelial Cells (HUVEC), we cannot rule out that the blunted induction of these genes is specific to this cell type rather than a general feature of endothelial cells. In agreement with this latter possibility, preliminary experiments showed a feeble induction of glycolytic genes in several, but not all, endothelial cells tested (data not shown).

The Hallmark subset of MSigDB contains signatures generated by a computational method based on the identification of overlaps across different gene sets and retaining those genes that display coordinate expression [24]. In spite of being an invaluable and widely used resource, our results suggests that the MSigDB hypoxia signature shows some shortcomings. For one thing, the signature lacks many hypoxia-regulated genes, containing only 37% of the genes strongly and consistently regulated by hypoxia across different cell types and experimental conditions (Figure 6A). In addition, the 114 core genes identified by the meta-analysis and not present in the Hallmark signature, include well characterized hypoxia-induced genes such as BCKDHA, EGLN1, several KDM family members and LOXL2 among others (Appendix A). On the other hand, the Hallmark signature includes some genes that are induced only in specific cell types or experimental conditions (Figure 6D) and thus, cannot be considered general hypoxia responsive genes. This result is particularly interesting as it explains the contradictory reports regarding the effect of hypoxia on specific genes such as HMOX1 [31,32,33,34,35] and PPARG1 [36,37,38,39,40,41,42].

In summary, herein we describe a formal meta-analysis approach that identifies the core transcriptional response to hypoxia. In addition to the identity of the genes, the approach results in a estimate of the magnitude of their change in expression in response to hypoxia. We also describe an approach to determine a minimum effect size to be used in combination with the statistical significance to identify biologically relevant changes in response to hypoxia.

## Figures and Tables

**Figure 1 biomedicines-10-02229-f001:**
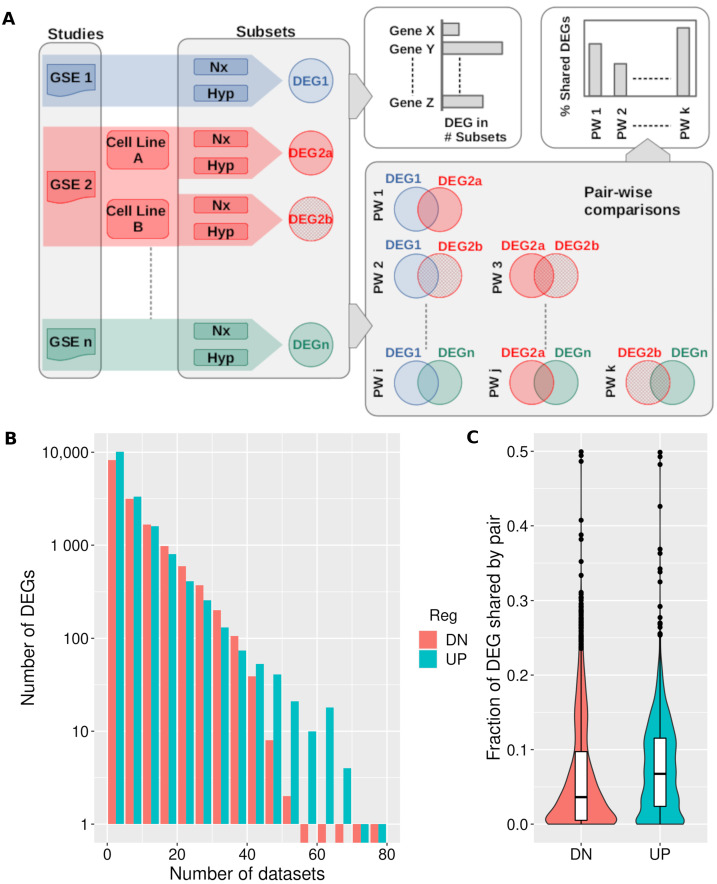
Hypoxic transcriptomes show limited overlap. (**A**) Diagram depicting the process used to compare hypoxic transcriptomes. Normoxic (Nx samples) and hypoxic (Hyp samples) replicates from the relevant studies (“GSE1”, “GSE2”, …“GSEn”) were processed to identify genes whose expression was significantly affected by hypoxia (Differentially Expressed Genes, “DEGs”). In those studies analyzing more than a single cell line, time of exposure to hypoxia, or oxygen tension, samples were grouped to generate homogeneous subsets and the effect of hypoxia on gene expression was analyzed in each individual subset. The figure represents this situation in the case of GSE2 (shaded in red color), an hypothetical study that analyzed the effect of hypoxia in two different cell types. The number of datasets were a gene was found to be a DEG was recorded (see panel B). In addition, pairwise comparisons (“PW1”, “PW2”, …“PWk”) between the 81 individual lists of DEGs were performed (lower right) to calculate the fraction of shared DEGs by each pair (see panel C). (**B**) Histogram showing the distribution of the number of down- (“DN”) or up-regulated (“UP”) genes binned by the number of datasets were the gene was found to be significantly regulated. In order to show the whole range, the y-axis is log10 scale. (**C**) Violin and overlaid boxplot showing the distribution of the fraction of down- (“DN”) or up-regulated (“UP”) genes shared in all 3240 possible pair-wise comparisons of the 81 datasets. For each pair of datasets A and B fraction of shared DEG was calculated as $\frac{\left|A\cap B\right|}{\left|A\cup B\right|}$.

**Figure 2 biomedicines-10-02229-f002:**
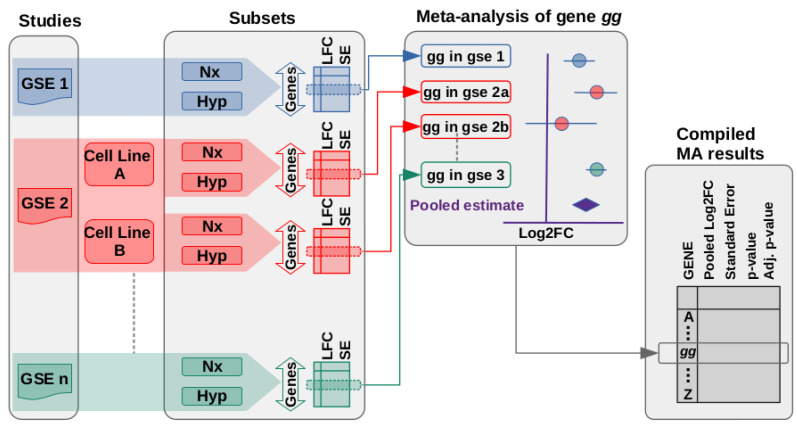
Integration of studies and gene-level meta-analysis. Normoxic (Nx samples) and hypoxic (Hyp samples) replicates from the relevant studies (“GSE1”, “GSE2”, …“GSEn”) were processed to produce a table recording the effect of hypoxia on the expression of each gene (Log_2_ fold-change, labeled as “LFC”) and the standard error associated to this estimation (“SE”). Complex studies were subdivided to produce minimal subsets of data (see Figure 1). Then, the results obtained for each individual gene (represented by gene “*gg*” in the figure) were integrated into a random-effects model meta-analysis to produce a pooled estimate of the effect of hypoxia on each gene. Note that a meta-analysis is performed for each individual gene. Finally, the results of the gene-level meta-analyses were integrated into a single list (“Compiled MA results”). *p*-values from individual meta-analyses were corrected for multiple-testing.

**Figure 3 biomedicines-10-02229-f003:**
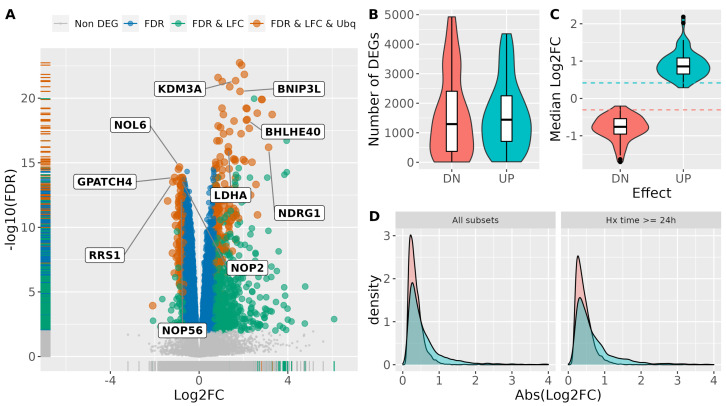
Identification of a common set of hypoxia-regulated genes. (**A**) The graph represents the pooled effect of hypoxia on gene expression (Log_2_FC hypoxia over normoxia) against the statistical significance of the effect (−log_10_
*FDR*−adjusted *p*−value) according to the meta-analysis. Genes are represented as dots and their color indicates the effect of hypoxia: grey, genes not regulated by hypoxia (*FDR*−adjusted *p*−value ≥0.01); blue, genes mildly affected by hypoxia (*FDR*−adjusted *p*−value <0.01 and $|Log_{2}FC|\leq 0.7$); green, genes strongly regulated by hypoxia (*FDR*−adjusted *p*−value <0.01 and $|Log_{2}FC|>0.7$), and dark red, ubiquitous genes regulated by hypoxia (strongly regulated genes present in > 90% of the subsets). Non significant genes are represented by smaller dots to avoid saturation. (**B**) Distribution of the number of genes found significantly (*FDR*−adjusted *p*−value ≥0.01) down- (“DN”) or up-regulated (“UP”) by hypoxia in individual studies. (**C**) Distribution of the median effect size (Log_2_FC Hypoxia over Normoxia) of hypoxia on repressed (“DN”) or induced (“UP”) genes in individual studies. The red and blue dotted lines correspond to the median effect size for repressed and induced genes respectively, according to the meta-analysis pooled estimates. (**D**) Distribution of Log_2_FC values for significantly down-regulated (blue) and up-regulated (red) genes, both in a meta-analysis including all the subsets, as well as in a meta-analysis restricted to subsets corresponding to hypoxia treatments of 24 h or more.

**Figure 4 biomedicines-10-02229-f004:**
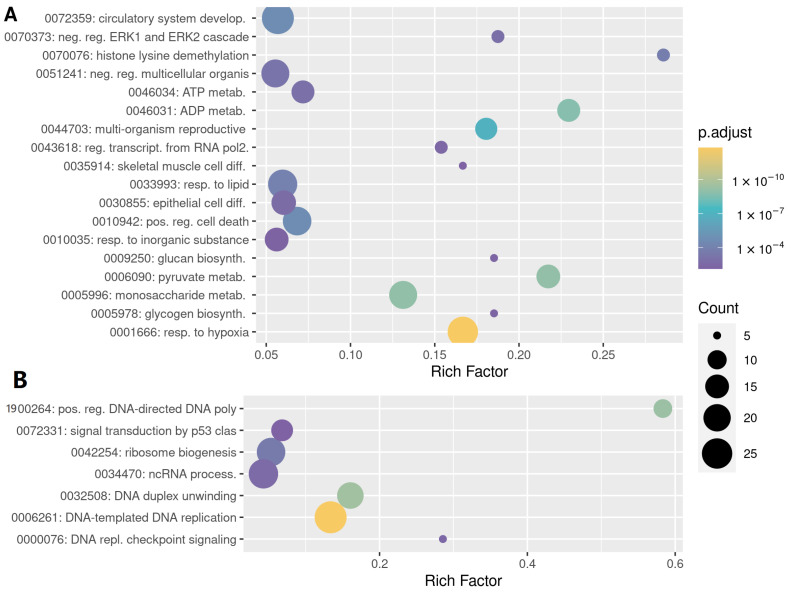
Functional enrichment analysis of the core hypoxic signature. Gene Ontology terms (Gene Ontology ID: Description) significantly enriched among the core of hypoxia up- (**A**) and down- (**B**) regulated genes. The analysis was restricted to terms of the “Biological process” domain. “Rich Factor”, represents the fraction of category genes regulated by hypoxia. Count, absolute number of category genes regulated by hypoxia.

**Figure 5 biomedicines-10-02229-f005:**
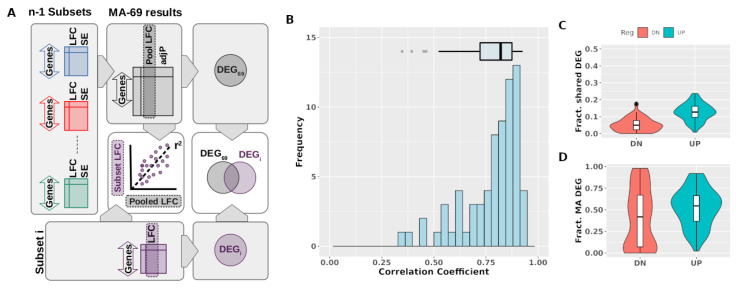
Meta-analyses accurately predict the identity of DEG and the magnitude of the change in their expression. (**A**) Diagram depicting the procedure to assess meta-analyses consistency using leave-one-out cross-validation. For each iteration, a meta-analysis spanning 69 subsets (“MA-69”) was performed, and then the concordance between expression changes in the individual subset left out and the pooled estimates in the MA-69 was calculated (central column in the diagram). In addition, the list of differentially expressed genes derived from the meta-analysis (“$DEG_{69}$”) and from the individual transcriptome (“$DEG_{i}$”) were compared to assessed their overlap (right column in the diagram). The whole procedure shown in panel A was repeated leaving out a different subset each time until all possibilities were exhausted. The distribution of the 70 resulting correlation coefficients is shown in panel B and the overlap between DEG lists in panels C and D. (**B**) Histogram and boxplot showing the distribution of the Pearson’s correlation coefficients between the pooled estimates and the actual effect size observed in each left-out subset. (**C**) Genes showing a statistically significant (FDR<0.01) change in expression were obtained from each meta-analysis ($DEG_{MA}$) and the corresponding left-out individual subset ($DEG_{i}$). The overlap between both lists of DEGs was calculated as $\frac{|DEG_{MA}\cap DEG_{i}|}{|DEG_{MA}\cup DEG_{i}|}$. The graph represents the distribution of the overlap values obtained in the 70 iterations of the process. To facilitate the comparison, the graph uses the same y-axis scale than the one in Figure 1C. (**D**) A list of “core” hypoxia responsive genes was derived from each meta-analyses as ubiquitous genes (expression detected in at least 90% of the datasets included in the meta-analyses) showing a robust (|LFC|>0.7) and significant (FDR<0.01) change in expression ($DEG_{core}$). This list was then compared the list of genes showing a significant (FDR<0.01) change in expression in the remaining subset ($DEG_{i}$). The proportion of “core” genes present in the DEG from the individual experiment was calculated as $\frac{|DEG_{core}\cap DEG_{i}|}{|DEG_{core}|}$. The graph represents the distribution of the values obtained in the 70 iterations of the process.

**Figure 6 biomedicines-10-02229-f006:**
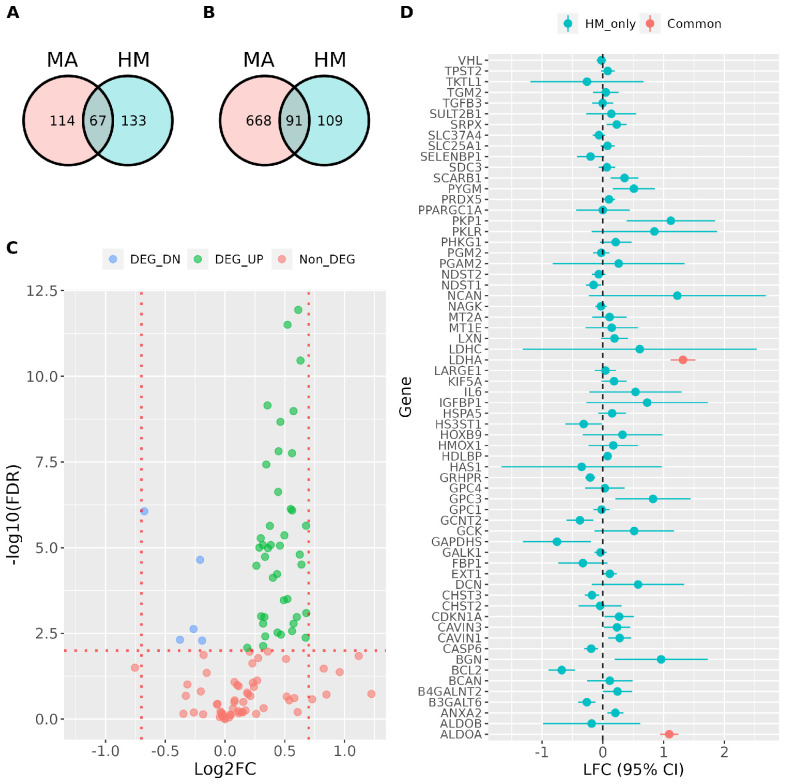
Comparison of Meta-analyses-derived and Hallmark hypoxic signatures. (**A**,**B**) Venn diagrams showing the number of genes shared by the Hallmark hypoxic signature (labeled HM) and the meta-analysis derived (labeled MA) core geneset (**A**), including genes expressed in over 90% of the studies, or an extended geneset not restricted to ubiquitously expressed genes (**B**). (**C**) Effect of hypoxia (pooled estimates from meta-analyses) on the 109 Hallmark hypoxia signature genes not present in the meta-analyses geneset (see panel B). Genes showing a statistically significant induction or repression by hypoxia are shown as green or blue dots respectively, while those whose expression is not significantly altered are shown as red dots. The red vertical dotted lines mark threshold effect size of 0.7 logarithmic units. The red horizontal dotted line marks the significance threshold (adjusted *p*−value < 0.01). (**D**) Forest plot showing the estimated effect of hypoxia on the expression of Hallmark genes not significantly up-regulated (those labelled in red and blue colors in panel C) together with the confidence interval for the point estimate. For comparison, the paralogs of ALDOB and LDHC, ALDOA and LDHA respectively, are included and labelled in red color.

**Table 1 biomedicines-10-02229-t001:** Datasets used in the meta-analysis. The number of independent GEO entries (Series IDs, GSE) that met the criteria described under methods is shown (“Studies”). For each study, only those samples corresponding to normoxic and hypoxic conditions were retrieved ignoring any other treatment that the study could have included. The total number of samples meeting these criteria is shown (“Samples”). In those cases were a study included more than one cell line, different degrees of hypoxia or different exposure times, the samples were divided into subsets including a single level for each one of these variables. The total number of subsets generated is shown (“Subsets”). The column “Cell lines” indicates the number of different cell lines included in the dataset. The rows contain values corresponding to the dataset prior (“Initial”) and after (“Filtered”) filtering to remove outlier studies.

Dataset	Studies	Samples	Subsets	Cell Lines
Initial	46	472	81	38
Filtered	43	430	70	34

## Data Availability

Data supporting reported results can be found at NCBI’s Gene Expression Ommibus (GEO) repository, Appendix A includes the ID for the studies used herein.

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
