# Peer review of "Formal Meta-Analysis of Hypoxic Gene Expression Profiles Reveals a Universal Gene Signature"

_biomedicines, 2022, doi:10.3390/biomedicines10092229_

Round 1
Reviewer 1 Report
The work of Puente-Santamaria et al used a formal meta-analysis in order to identify a universal signature of hypoxia-responsive genes. The article is well written and quite innovative on its approach. It identified interesting features of cell response to hypoxia and pointed out that endothelial cells may behave differently than other cell types.
Comments
- The work focused on gene expression changes assessed in cell lines in vitro. This should be clearly stated in the summary.
- In the exclusion-inclusion criteria, it should be mentioned if only human cell lines were considered or if cells from other species were included in the study. Similarly, it should also be mentioned whether cycling/intermittent hypoxia was included or excluded.
- Part of the genes that are downregulated are downregulated because of the decrease in energy availability within cells upon hypoxia exposure and not because of a specific regulation. This should be discussed in the article.
Author Response
Response to Reviewer 1 Comments
Point 1: The work focused on gene expression changes assessed in cell lines in vitro. This should be clearly stated in the summary.
Response 1: we agree with the reviewer that this is an important aspect that should be mentioned in the summary. Thus, we have modified two sentences of the summary to include it:
“We applied a formal meta-analysis procedure to a dataset comprising 430 RNA-seq samples from 43 individual studies including 34 different cell types, to derive a pooled estimate of the effect of hypoxia on gene expression in human cell lines grown in vitro.”
“Overall, we have developed a robust gene signature that characterizes the transcriptomic response of human cell lines exposed to hypoxia in vitro by applying a formal meta-analysis to gene expression profiles.”
Point 2: In the exclusion-inclusion criteria, it should be mentioned if only human cell lines were considered or if cells from other species were included in the study. Similarly, it should also be mentioned whether cycling/intermittent hypoxia was included or excluded.
Response 2: To clarify the inclusion/exclusion criteria, we have included the following sentence in the relevant section of materials and methods:
“We also excluded studies that used cycling/intermittent hypoxia or that were performed in non-human cell lines”
In addition, we have modified the first sentence of this section to include the adjective “sustained” to stress that the study only focused on continuous hypoxia exposure: “The meta-analysis intended to identify the effect of sustained hypoxia on early gene expression in human cells compared to normoxic controls.”
Point 3: Part of the genes that are downregulated are downregulated because of the decrease in energy availability within cells upon hypoxia exposure and not because of a specific regulation. This should be discussed in the article.
Response 3: We agree that gene repression could be, at least in part, due to indirect mechanisms. To acknowledge this possibility we have included the following paragraph in the discussion:
“The effect of hypoxia on gene repression is apparently weaker than on gene induction, as indicated by relative smaller number of repressed genes above the effect size threshold. This could be a consequence of repression being indirectly regulated by HIFs (pmid19386601,pmid19255431,pmid20061373) and would explain the higher heterogeneity of the response observed for repressed genes (figure 1). The indirect regulation could occur via transcriptional regulators acting downstream of HIF (pmid29690561) or be a consequence of the cellular adaptations to hypoxia. For example, alteration of energy availability during hypoxia could prevent cell division with the concomitant down-regulation of genes involved in DNA synthesis and cell cycle progression. Thus, further work is required fully understand the mechanisms responsible for gene repression during hypoxia.”
Reviewer 2 Report
The authors have conducted a very thorough meta-analysis of the hypoxic gene expression profiles and it was very interesting that large proportion of the transcriptome is significantly regulated by hypoxia.
Minor grammar edits could be done. Example-
Integrating transcriptional profiles results in identifying gene expression signatures that are more robust than those obtained for individual datasets. However, a direct comparison of datasets derived from heterogeneous experimental conditions is impossible, and their integration requires applying specific meta-analysis techniques. The transcriptional response to hypoxia has been the focus of intense research due to its central role in tissue homeostasis and prevalent diseases. Accordingly, many studies have determined the gene expression profile of hypoxic cells. Yet, despite this wealth of information, little effort has been made to integrate these datasets to produce a robust hypoxic signature. We applied a formal meta-analysis procedure to datasets comprising 425 RNA-seq samples from 42 individual studies, including 33 different cell types, to derive a pooled estimate of the effect of hypoxia on gene expression. This approach 10 revealed that a large proportion of the transcriptome is significantly regulated by hypoxia (8556 out of 11 20888 genes identified across studies). However, only a tiny fraction of the differentially expressed genes (1265 genes, 15%) show an effect size that, according to comparisons to gene pathways known to be regulated by hypoxia, is likely to be biologically relevant. By focusing on genes ubiquitously expressed, we identified a signature of 291 genes robustly and consistently regulated by hypoxia. We have developed a robust gene signature that characterizes the transcriptomic response to low oxygen by applying a formal meta-analysis to hypoxic gene profiles.
Also, adding other types of ontology analysis would be helpful to the research. For example, pathway enrichment GO analysis would help suggest what all pathways are enriched in this study.
Author Response
Response to Reviewer 2 Comments
Point 1: Minor grammar edits could be done. Example-
Integrating transcriptional profiles results in identifying gene expression signatures that are more robust than those obtained for individual datasets. However, a direct comparison of datasets derived from heterogeneous experimental conditions is impossible, and their integration requires applying specific meta-analysis techniques. The transcriptional response to hypoxia has been the focus of intense research due to its central role in tissue homeostasis and prevalent diseases. Accordingly, many studies have determined the gene expression profile of hypoxic cells. Yet, despite this wealth of information, little effort has been made to integrate these datasets to produce a robust hypoxic signature. We applied a formal meta-analysis procedure to datasets comprising 425 RNA-seq samples from 42 individual studies, including 33 different cell types, to derive a pooled estimate of the effect of hypoxia on gene expression. This approach 10 revealed that a large proportion of the transcriptome is significantly regulated by hypoxia (8556 out of 11 20888 genes identified across studies). However, only a tiny fraction of the differentially expressed genes (1265 genes, 15%) show an effect size that, according to comparisons to gene pathways known to be regulated by hypoxia, is likely to be biologically relevant. By focusing on genes ubiquitously expressed, we identified a signature of 291 genes robustly and consistently regulated by hypoxia. We have developed a robust gene signature that characterizes the transcriptomic response to low oxygen by applying a formal meta-analysis to hypoxic gene profiles.
Response 1: We thank the reviewer for providing us a revised version of the summary. We have included al the suggested changes to it and revised the remaining text of the manuscript .
Point 2: Also, adding other types of ontology analysis would be helpful to the research. For example, pathway enrichment GO analysis would help suggest what all pathways are enriched in this study.
Response 2: Gene Ontology (GO) is a formal annotation of gene function according to three domains: molecular function, cellular component and biological process. Although there is not a “pathway” domain in GO, the closest would be the “biological process”. Figure 4 of the manuscript described the enrichment of GO terms from the biological process domain. We have included a sentence in the figure legend and in the relevant section of materials and methods to specify that the GO analysis was restricted to biological process terms:
“To focus broader biological programs rather than molecular activities, the analysis was restricted to terms of the Biological Process domain”
To interrogate proper pathways databases we performed Gene Set Enrichment Analysis (GSEA) against the Biocarta, KEGG, PID, REACTOME, and Wikipathways subsets of the MsigDB. In agreement with GO enrichment analysis, GSEA indicates that hypoxia-induced genes are associated with the HIF pathway and glucose metabolism whereas repressed genes are mainly associated with DNA replication, cell cycle progression. The results of these analyses have been included in the revised version of the manuscript as new supplementary tables A6 and A7 and are discussed in the main text.